# A Neural Network MCMC Sampler That Maximizes Proposal Entropy

**DOI:** 10.3390/e23030269

**Published:** 2021-02-25

**Authors:** Zengyi Li, Yubei Chen, Friedrich T. Sommer

**Affiliations:** 1Redwood Center for Theoretical Neuroscience, Berkeley, CA 94720, USA; yubeic@berkeley.edu (Y.C.); fsommer@berkeley.edu (F.T.S.); 2Department of Physics, University of California Berkeley, Berkeley, CA 94720, USA; 3Berkeley AI Research, University of California Berkeley, Berkeley, CA 94720, USA; 4Helen Wills Neuroscience Institute, University of California Berkeley, Berkeley, CA 94720, USA; 5Neuromorphic Computing Group, Intel Labs, 2200 Mission College Blvd., Santa Clara, CA 95054-1549, USA

**Keywords:** MCMC, neural network sampler, maximum entropy, energy-based model

## Abstract

Markov Chain Monte Carlo (MCMC) methods sample from unnormalized probability distributions and offer guarantees of exact sampling. However, in the continuous case, unfavorable geometry of the target distribution can greatly limit the efficiency of MCMC methods. Augmenting samplers with neural networks can potentially improve their efficiency. Previous neural network-based samplers were trained with objectives that either did not explicitly encourage exploration, or contained a term that encouraged exploration but only for well structured distributions. Here we propose to maximize proposal entropy for adapting the proposal to distributions of any shape. To optimize proposal entropy directly, we devised a neural network MCMC sampler that has a flexible and tractable proposal distribution. Specifically, our network architecture utilizes the gradient of the target distribution for generating proposals. Our model achieved significantly higher efficiency than previous neural network MCMC techniques in a variety of sampling tasks, sometimes by more than an order magnitude. Further, the sampler was demonstrated through the training of a convergent energy-based model of natural images. The adaptive sampler achieved unbiased sampling with significantly higher proposal entropy than a Langevin dynamics sample. The trained sampler also achieved better sample quality.

## 1. Introduction

Sampling from unnormalized distributions is important for many applications, including statistics, simulations of physical systems and machine learning. However, the inefficiency of state-of-the-art sampling methods remains a main bottleneck for many challenging applications, such as protein folding [1] and energy-based model training [2].

A prominent strategy for sampling is the Markov Chain Monte Carlo (MCMC) method [3]. In MCMC, one chooses a transition kernel that leaves the target distribution invariant and constructs a Markov Chain by applying the kernel repeatedly. The MCMC method relies only on the ergodicity assumption. Other than that, it is general: if enough computation is performed, the Markov Chain generates correct samples from any target distribution, no matter how complex the distribution is. However, the performance of MCMC depends critically on how well the chosen transition kernel explores the state space of the problem. If exploration is ineffective, samples will be highly correlated and of very limited use for downstream applications. Thus, despite the theoretical guarantee that MCMC algorithms are exact, practically they may still suffer from inefficiencies.

Take, for example, Hamiltonian Monte Carlo (HMC) sampling [4], a type of MCMC technique. HMC is regarded as the state-of-the-art for sampling in continuous spaces [5]. It uses a set of auxiliary momentum variables and generates new samples by simulating Hamiltonian dynamics starting from the previous sample. This allows the sample to travel in state space much further than possible with other techniques, most of whom have more pronounced random walk behavior. Theoretical analysis shows that the cost of traversing in *d*-dimensional state space and generating an uncorrelated proposal is $O(d^{\frac{1}{4}})$ for HMC, which is lower than $O(d^{\frac{1}{3}})$ for Langevin Monte Carlo, and O(d) for random walk [6]. However, unfavorable geometry of a target distribution may still render HMC ineffective because the Hamiltonian dynamics have to be simulated numerically. Numerical errors in the simulation are corrected by a Metropolis–Hastings (MH) accept–reject step of a proposed sample. If the the target distribution has unfavorable geometric properties, for example, large differences in variance along different directions, the numerical integrator in HMC will produce high errors, leading to a very low accept probability [7]. For simple distributions, this inefficiency can be mitigated by an adaptive re-scaling matrix [4]. For analytically tractable distributions, one can also use the Riemann manifold HMC method [8]. However, in most other cases, the Hessian required in the Riemann manifold HMC algorithm is often intractable or expensive to compute, preventing its application.

Recently, approaches have been proposed that inherit the exact sampling property from the MCMC method, while potentially mitigating the described issues of unfavorable geometry. One approach is MCMC samplers augmented with neural networks [9,10,11]; the other approach is neural transport MCMC techniques [12,13]. A disadvantage of these recent techniques is that their objectives optimize the quality of proposed samples, but do not explicitly encourage the exploration speed of the sampler. One notable exception is L2HMC [10], a method whose objective includes the size of the expected L2 jump, thereby encouraging exploration. However, the L2 expected jump objective is not very general; it only works for simple distributions (see Figure 1, and below).

In another recent work [14], exploration speed was encouraged by a quite general objective: the entropy of the proposal distribution. In continuous space, the entropy of a distribution is essentially the logarithm of its volume in state space. Thus, the entropy objective naturally encourages the proposal distribution to “fill up” the target state space and possible independent of the geometry of the target distribution. The authors demonstrated the effectiveness of this objective on samplers with simple linear adaptive parameters.

We highlight the difference between L2 expected jump objective and entropy objective in Figure 1. A neural network sampler, trained with the entropy-based objective, generates samples that explore the target distribution quite well. In contrast, sampling by constructing proposals with higher expected L2 jumps leads to a less desirable result (right panel).

In the paper we make the following contributions:1.Here, we employed the entropy-based objective in a neural network MCMC sampler for optimizing exploration speed. To build the model, we designed a novel flexible proposal distribution wherein the optimization of the entropy objective is tractable.2.Inspired by the HMC and the L2HMC [10] algorithm, the proposed sampler uses a special architecture that utilizes the gradient of the target distribution to aid sampling.3.We demonstrate a significant improvement in sampling efficiency over previous techniques, sometimes by an order of magnitude. We also demonstrate energy-based model training with the proposed sampler and demonstrate higher exploration and higher resultant sample quality.

The reminder of the paper is organized as follows. Section 2 briefly introduces MCMC methods. In Section 3 we formulate the model. Section 4 discusses relationships and differences of the new model and HMC-based models from the literature. In Section 5, we provide experimental results. The paper concludes with a discussion in Section 6.

## 2. Preliminaries: MCMC Methods from Vanilla to Learned

Consider the problem of sampling from a target distribution $p\left(x\right)=e^{-U(x)}/Z$ defined by the energy function U(x) in a continuous state space. MCMC methods solve this problem by constructing and running a Markov Chain with a transition probability $p(x^{\prime }|x)$ that leaves p(x) invariant. The most general invariance condition is: $p\left(x^{\prime }\right)=\int p\left(x^{\prime }|x\right)p\left(x\right)dx$ for all $x^{\prime }$, which is typically enforced by the simpler but more restrictive condition of detailed balance: $p\left(x\right)p\left(x^{\prime }|x\right)=p\left(x^{\prime }\right)p\left(x|x^{\prime }\right)$.

For a general distribution p(x) it is difficult to directly construct a $p(x^{\prime }|x)$ that satisfies detailed balance. However, one can easily (Up to ergodic and aperiodic assumptions ) make any transition probability satisfy detailed balance by adding a Metropolis–Hastings (M–H) accept–reject step [15]. When we sample $x^{\prime }$ at step *t* from an arbitrary proposal distribution $q(x^{\prime }|x^{t})$, the M–H accept–reject process accepts the new sample $x^{t+1}=x^{\prime }$ with probability
(1)$$A\left(x^{\prime },x\right)=\min\left(1,\frac{p\left(x^{\prime }\right)q\left(x^{t}|x^{\prime }\right)}{p\left(x^{t}\right)q\left(x^{\prime }|x^{t}\right)}\right).$$

If $x^{\prime }$ is rejected, the new sample is set to the previous state $x^{t+1}=x^{t}$. This transition kernel $p(x^{\prime }|x)$ constructed from $q(x^{\prime }|x)$ and $A(x^{\prime },x)$ leaves any target distribution p(x) invariant.

Most popular MCMC techniques use the described M–H accept–reject step to enforce detailed balance, for example, the random walk Metropolis (RWM), the metropolis-adjusted Langevin algorithm (MALA) and the Hamiltonian Monte Carlo (HMC). For brevity, we will focus on MCMC methods that use the M–H step, although some alternatives do exist [16]. These methods share the requirement that the accept probability in the M–H step (Equation (1)) must be tractable to compute. For two of the mentioned MCMC methods this is indeed the case. In the Gaussian random-walk sampler, the proposal distribution is a Gaussian around the current position: $x^{\prime }=x+\epsilon *N\left(0,I\right)$, which has the form $x^{\prime }=x+z$. Thus, forward and reverse proposal probabilities are given by $q\left(x^{\prime }|x\right)=p_{N}\left[(x^{\prime }-x)/\epsilon \right]$ and $q\left(x|x^{\prime }\right)=p_{N}\left[-(x^{\prime }-x)/\epsilon \right]$, where $p_{N}$ denotes the density function of a Gaussian with a 0 mean and unit diagonal variance. The probability ratio $\frac{q(x^{t}|x^{\prime })}{q(x^{\prime }|x^{t})}$ used in the M–H step (Equation (1)) is therefore equal to 1. In MALA the proposal distribution is a single step of Langevin dynamics with step size ϵ: $x^{\prime }=x+z$ with $z=-\frac{\epsilon^{2}}{2}\partial_{x}U\left(x\right)+\epsilon N\left(0,I\right)$. We then have $q\left(x^{\prime }|x\right)=p_{N}\left[\left(x^{\prime }-x\right)/\epsilon +\frac{\epsilon }{2}\partial_{x}U\left(x\right)\right]$ and $q\left(x|x^{\prime }\right)=p_{N}\left[-\left(x^{\prime }-x\right)/\epsilon +\frac{\epsilon }{2}\partial_{x^{\prime }}U\left(x^{\prime }\right)\right]$. Both the forward and reverse proposal probabilities are tractable since they are the densities of Gaussians evaluated at a known location.

Next we introduce the HMC sampler and show how it can be formulated as an M–H sampler. Basic HMC involves a Gaussian auxiliary variable *v* of the same dimension as *x*, which plays the role of the momentum in physics. HMC sampling consists of two steps: (1) The momentum is sampled from a normal distribution N(v;0,I). (2) The Hamiltonian dynamics are simulated for a certain duration with initial condition *x* and *v*, typically by running a few steps of the leapfrog integrator [4]. Then, an M–H accept–reject process with accept probability $A\left(x^{\prime },v^{\prime },x,v\right)=\min\left(1,\frac{p\left(x^{\prime },v^{\prime }\right)q\left(x,v|x^{\prime },v^{\prime }\right)}{p\left(x,v\right)q\left(x^{\prime },v^{\prime }|x,v\right)}\right)=\min\left(1,\frac{p\left(x^{\prime }\right)p_{N}\left(v^{\prime }\right)}{p\left(x\right)p_{N}\left(v\right)}\right)$ is performed to correct for error in the integration process. We have $\frac{q(x,v|x^{\prime },v^{\prime })}{q(x^{\prime },v^{\prime }|x,v)}=1$ since the Hamiltonian transition is volume-preserving over (x,v). Both HMC steps leave the joint distribution p(x,v) invariant; therefore, HMC samples form the correct distribution p(x) after marginalizing over *v*. To express basic HMC in the standard M–H scheme, steps 1 and 2 can be aggregated into a single proposal distribution on *x* with the proposal probability: $q\left(x^{\prime }|x\right)=p_{N}\left(v\right)$ and $q\left(x|x^{\prime }\right)=p_{N}\left(v^{\prime }\right)$. Note, although the probability $q(x^{\prime }|x)$ can be calculated after the Hamiltonian dynamics are simulated, this term is intractable for general *x* and $x^{\prime }$. The reason is that it is difficult to solve for the *v* at *x* to make the transition to $x^{\prime }$ using the Hamiltonian dynamics. This issue is absent in RWM and MALA, where $q(x^{\prime }|x)$ is tractable for any *x* and $x^{\prime }$.

Previous work on augmenting MCMC sampler with neural networks also relied on the M–H procedure to ensure asymptotic correctness of the sampling process, for example [9,10]. They used HMC style accept–reject probabilities that lead to intractable $q(x^{\prime }|x)$. Here, we strive for a flexible sampler for which $q(x^{\prime }|x)$ is tractable. This maintains the tractable M–H step while allowing us to train this sampler to explore the state space by directly optimizing the proposal entropy objective, which is a function of $q(x^{\prime }|x)$.

## 3. A Gradient-Based Sampler with Tractable Proposal Probability

We “abuse” the power of neural networks to design a sampler that is flexible and has tractable proposal probability $q(x^{\prime }|x)$ between any two points. However, without any extra help, the sampler would be modeling a conditional distribution $q(x^{\prime }|x)$ with brute force, which might be possible but requires a large model capacity. Thus, our method uses the gradient of the target distribution to guide proposal distribtion. Specifically, we use an architecture similar to L2HMC [10], which itself was inspired by the HMC algorithm and RealNVP [17]. To quantify the benefit of using the target distribution gradient, we provide ablation studies of our model in the Section A.1.

### 3.1. Proposal Model and How to Use Gradient Information

We restrict our sampler to the simple general form $x^{\prime }=x+z$. As discussed in Section 2, the sampler will have tractable proposal probability if one can calculate the probability of any given *z*. To fulfill this requirement, we model vector *z* by a flow model (For more details on flow models, see [18,19]): $z=f(z_{0};x,U)$, with inverse $z_{0}=f^{-1}\left(z;x,U\right)$. Here $z_{0}$ is sampled from a fixed Gaussian base distribution. The flow model *f* is a flexible and trainable invertible function of *z* conditioned on x,U, and it has tractable Jacobian determinant w.r.t. *z*. The flow model *f* can be viewed as a change of variable from the Gaussian base distribution z0 to *z*. The proposed sampler then has tractable forward and reverse proposal probabilities: $q\left(x^{\prime }|x\right)=p_{Z}\left(x^{\prime }-x;x\right)$, $q\left(x|x^{\prime }\right)=p_{Z}\left(x-x^{\prime };x^{\prime }\right)$, where $p_{Z}\left(z;x\right)=p_{N}\left(z_{0}\right){\left|\frac{\partial z}{\partial z_{0}}\right|}^{-1}$ is the density defined by the flow model *f*. Note, this sampler is ergodic and aperiodic, since $q(x^{\prime }|x)\neq 0$ for any *x* and $x^{\prime }$, which follows from the invertibility of *f*. Thus, combined with the M–H step, the sampling process will be asymptotically correct. The sampling process first consists of drawing from $p_{N}\left(z_{0}\right)$ and then evaluating $z=f(z_{0};x,U)$ and $q(x^{\prime }|x)$. Next, the reverse $z_{0}^{\prime }=f^{-1}\left(-z;x+z,U\right)$ is evaluated at $x^{\prime }=x+z$ to obtain the reverse proposal probability $q(x|x^{\prime })$. Finally, the sample is accepted with the standard M–H rule.

For the flow model *f*, we use an architecture similar to a non-volume preserving coupling-based flow RealNVP [17]. In the coupling-based flow, half of the components of the state vector are kept fixed and used to update the other half through an affine transform parameterized by a neural network. The gradient of the target distribution enters our model in those affine transformations. To motivate this particular model choice, we take a closer look at the standard HMC algorithm. Basic HMC starts with drawing a random initial momentum $v^{0}$, followed by several steps of leapfrog integration. In the *n*th leapfrog step, the integrator first updates *v* with a half step of the gradient: $v^{n^{\prime }}=v^{n-1}-\frac{\epsilon }{2}\partial_{x}U\left(x^{n-1}\right)$, followed by a full step of *x* update: $x^{n}=x^{n-1}+\epsilon v^{n^{\prime }}$ and another half step of *v* update: $v^{n}=v^{n^{\prime }}-\frac{\epsilon }{2}\partial_{x}U\left(x^{n}\right)$. After several steps, the overall update of *x* can be written as: $x^{n}=x^{0}+\sum_{i=0}^{n}v^{i^{\prime }}$, which has the form $x^{\prime }=x+z$ with $z=\sum_{i}^{n}v^{i^{\prime }}=-nv^{0}-\frac{n\epsilon }{2}\left[\partial_{x}U\left(x^{0}\right)\right]-\epsilon \left[\sum_{i=1}^{n}\left(n-i\right)\partial_{x}U\left(x^{i}\right)\right]$. This equation suggests that when generating *z* through affine transformations, gradient should enter through the shift term with a negative sign.

### 3.2. Model Formulation

To formulate our model (Equations (2) and (3)), we use a binary mask vector *m* and its complement $\bar{m}$ to select half of *z*’s dimensions for update at a time. As discussed above, we include the gradient term with a negative sign in the shift term of the affine transform. We also use an element-wise scaling on the gradient term as in [10]. However, two issues remain. First, as required by the coupling-based architecture, the gradient term can only depend on the masked version of vector *z*. Second, it is unclear where the gradient should be evaluated to sample effectively. As discussed above, the sampler should evaluate the gradient at points far away from *x*, similar as in HMC, to travel long distances in the state space. To handle these issues, we use another neural network *R* which receives *x* and the masked *z* as input, and evaluates the gradient at x+R. During training, *R* learns where the gradient should be evaluated based on the masked *z*.

We denote the input to network *R* by $\zeta_{m}^{n}=\left(x,m⊙z^{n}\right)$ and the input to the other networks by $\xi_{m}^{n}=\left(x,m⊙z^{n},\partial U\left(x+R\left(\zeta_{m}^{n}\right)\right)\right)$, where ⊙ is the Hadamard product (element wise multiply). Further, we denote the neural network outputs that parameterize the affine transform by $S(\xi_{m}^{n})$, $Q(\xi_{m}^{n})$ and $T(\xi_{m}^{n})$, where exp[S] and *T* parameterize the element-wise scaling term and shift term in the affine transform, and exp[Q] gives the element-wise scaling term for the gradient. For notation clarity we omit dependencies of the mask *m* and all neural network terms on the step number *n*.

Additionally, we introduce a scale parameter ϵ, which modifies the *x* update to $x^{\prime }=x+\epsilon z$. We also define $\epsilon^{\prime }=\epsilon /\left(2N\right)$, with *N* the total number of *z* update steps. This parameterization makes our sampler equivalent to the MALA algorithm with step size ϵ at initialization, where the neural network outputs are zero. The resulting update rule is: (2)$$\begin{array}{cc} \;z^{n^{\prime }}\; & =\;m\;⊙z^{n-1}\;+\bar{m}⊙\left(z^{n-1}\;⊙\exp\left[S\left(\xi_{m}^{n-1}\right)\right]\;-\epsilon^{\prime }\left\{\partial U\left[x+R\left(\zeta_{m}^{n-1}\right)\right]\;⊙\;\exp\left[Q\left(\xi_{m}^{n-1}\right)\right]+T\left(\xi_{m}^{n-1}\right)\right\}\right) \end{array}$$(3)$$\begin{array}{cc} \;z^{n} & =\bar{m}⊙z^{n^{\prime }}+m⊙\left(z^{n^{\prime }}\;⊙\exp\left[S\left(\xi_{\bar{m}}^{n^{\prime }}\right)\right]-\epsilon^{\prime }\left\{\partial U\left[x+R(\zeta_{\bar{m}}^{n^{\prime }})\right]⊙\exp\left[Q\left(\xi_{\bar{m}}^{n^{\prime }}\right)\right]+T\left(\xi_{\bar{m}}^{n^{\prime }}\right)\right\}\right) \end{array}$$

The log determinant of *N* steps of transformation is:(4)$$\log\left|\frac{\partial x^{\prime }}{\partial z^{0}}\right|=\epsilon 1*1+\sum_{n=1}^{N}1*\left[\bar{m}⊙S\left(\xi_{m}^{n-1}\right)\right]+1*\left[m⊙S(\xi_{\bar{m}}^{n^{\prime }})\right]$$
where 1 is the vector of 1-entries with the same dimension as *z*, * denotes the dot product. This follows from the simple fact that the log determinant of affine transform $z^{\prime }=z⊙\exp\left(S\right)+T$ is 1*S.

### 3.3. Optimizing the Proposal Entropy Objective

The proposal entropy can be expressed as: (5)$$\;H\left(X^{\prime }|X=x\right)\;=\;-\;\int dx^{\prime }q\left(x^{\prime }|x\right)\log\left[q\left(x^{\prime }|x\right)\right]\;=\;-\;\int \;dz^{0}p_{N}\left(z^{0}\right)\left[\log\left(p_{N}\left(z^{0}\right)\right)\;-\;\log\left|\frac{\partial z^{N}}{\partial z^{0}}\right|\right]\;$$

For each *x*, we aim to optimize $S\left(x\right)=\exp\left[\beta H(X^{\prime }|X=x)\right]\times a\left(x\right)$, where $a\left(x\right)=\int A\left(x^{\prime },x\right)q\left(x^{\prime }|x\right)dx^{\prime }$ is the average accept probability of the proposal distribution at *x*. Following [14], we transform this objective into log space and use Jensen’s inequality to obtain a lower bound:$$\begin{array}{cc} \log S(x) & =\log\int A\left(x^{\prime },x\right)q\left(x^{\prime }|x\right)dx^{\prime }+\beta H\left(X^{\prime }|X=x\right)\\ \; & \geq \int \log\left[A\left(x^{\prime }x\right)\right]q\left(x^{\prime }|x\right)dx^{\prime }+\beta H\left(X^{\prime }|X=x\right)=L\left(x\right) \end{array}$$

The distribution $q(x^{\prime }|x)$ is reparameterizable; therefore, the expectation over $q(x^{\prime }|x)$ can be expressed as expectation over $p_{N}\left(z_{0}\right)$. By expanding the lower bound L(x) and omitting the (constant) entropy of the base distribution $p_{N}\left(z_{0}\right)$, we arrive at:(6)$$L\left(x\right)=\int dz^{0}p_{N}\left(z^{0}\right)\left[\min\left(0,\log\frac{p(x^{\prime })}{p(x)}+\log\frac{q(x|x^{\prime })}{q(x^{\prime }|x)}\right)-\beta \log\left|\frac{\partial x^{\prime }}{\partial z^{0}}\right|\right]$$

During training we maximize L(x) with *x* sampled from the target distribution p(x) if it is available, or with *x* obtained from the bootstrapping process [9] which maintains a buffer of samples and updates them continuously. Typically, only one sample of $z^{0}$ is used for each *x*.

A curious feature of our model is that during training one has to back-propagate over the gradient of the target distribution multiple times to optimize *R*. In [14] the authors avoid multiple back-propagation by stopping the derivative calculation at the density gradient term. In our experiment we did not use this trick and performed full back-propagation without encountering any issue. We found that stopping the derivative computation instead harms performance.

The entropy-based exploration objective contains a parameter β that controls the balance between acceptance rate and proposal entropy. As in [14], we use a simple adaptive scheme to adjust β to maintain a constant accept rate close to a target accept rate. The target accept rate is chosen empirically. As expected, we find that the target accept rate needs to be lower for more complicated distributions.

## 4. Related Work: Other Samplers Inspired by HMC

Here we discuss other neural network MCMC samplers and how they differ from our method. Methods we compare ours to in the results are marked with bold font.

**A-NICE-MC** [9], which was generalized in [20], used the same accept probability as HMC, but replaced the Hamiltonian dynamics with a flexible volume-preserving flow [21]. A-NICE-MC matches samples from $q(x^{\prime }|x)$ directly to samples from p(x), using adversarial loss. This permits training the sampler on empirical distributions, i.e., in cases where only samples, but not the density function, are available. The problem with this method is that samples from the resulting sampler can be highly correlated because the adversarial objective only optimizes for the quality of the proposed sample. If the sampler produces a high quality sample *x*, the learning objective does not encourage the next sample $x^{\prime }$ to be substantially different from *x*. The authors used a pairwise discriminator that empirically mitigated this issue but the benefit in exploration speed is limited.

Another related sampling approach is **neural transport MCMC** [12,13,22], which fits a distribution defined by a flow model $p_{g}\left(x\right)$ to the target distribution using $\mathrm{KL}[p_{g}\left(x\right)||p\left(x\right)]$. Sampling is then performed with HMC in the latent space of the flow model. Due to the invariance of the KL-divergence with respect to a change of variables, the “transported distribution” in *z* space $p_{g^{-1}}\left(z\right)$ will be fitted to resemble the Gaussian prior $p_{N}\left(z\right)$. Samples of *x* can then be obtained by passing *z* through the transport map. Neural transport MCMC improves sampling efficiency compared to sampling in the original space because a distribution closer to a Gaussian is easier to sample. However, the sampling cost is not a monotonic function of the KL-divergence used to optimize the transport map [23].

Another line of work connects the MCMC method to Variational Inference [24,25,26,27]. Simply put, they improve the variational approximation by running several steps of MCMC transitions initialized from a variational distribution. The MCMC steps are optimized by minimizing the KL-divergence between the resulting distribution and the true posterior. This amounts to optimizing a “burn in” process in MCMC. In our setup however, the exact sampling is guaranteed by the M–H process, thus the KL divergence loss is no longer applicable. Like in variational inference, the **normalizing flow Langevin MC** (NFLMC) [11] also used a KL divergence loss. Strictly speaking, this model is a normalizing flow but not an MCMC method. We compare our method to it, because the model architecture, like ours, uses the gradient of the target distribution.

Another related technique is [28], where the authors trained an independent M–H sampler by minimizing $\mathrm{KL}\left[p\left(x\right)q\left(x^{\prime }|x\right)\left|\right|p\left(x^{\prime }\right)q\left(x|x^{\prime }\right)\right]$. This objective can be viewed as a lower bound of the M–H accept rate. However, as discussed in [14], this type of objective is not applicable for samplers that condition on the previous state.

All the mentioned techniques have in common that their objective does not encourage exploration speed. In contrast, **L2HMC** [10,29] does encourage fast exploration of the state space by employing a variant of the expected square jump objective [30]: $L\left(x\right)=\int dx^{\prime }q\left(x^{\prime }|x\right)A\left(x^{\prime },x\right)\left|\right|x^{\prime }-x{\left|\right|}^{2}$. This objective provides a learning signal even when *x* is drawn from the exact target distribution p(x). L2HMC generalized the Hamiltonian dynamics with a flexible non-volume-preserving transformation [17]. The architecture of L2HMC is very flexible and uses gradient of target distribution. However, the L2 expected jump objective in L2HMC improves exploration speed only in well-structured distributions (see Figure 1).

The shortcomings of the discussed methods led us to consider the use of an entropy-based objective. However, L2HMC does not have tractable proposal probability $p(x^{\prime }|x)$, preventing the direct application of the entropy-based objective. In principle, the proposal entropy objective could be optimized for the L2HMC sampler with variational inference [31,32], but our preliminary experiments using this idea were not promising. Therefore, we designed our sampler to possess tractable proposal probability and investigated tractable optimization of the proposal entropy objective.

## 5. Experimental Result

### 5.1. Synthetic Dataset and Bayesian Logistic Regression

First we demonstrate that our technique accelerates sampling of the funnel distribution, a particularly challenging example from [33]. We then compare our model with A-NICE-MC [9], L2HMC [10], normalizing flow Langevin MC (NFLMC) [11] and NeuTra [12] on several other synthetic datasets and a Bayesian logistic regression task. We additionally compare to gradMALA [14] to show the benefit of using neural network over linear adaptive sampler. For all experiments, we report effective sample size [34] per M–H step (ESS/MH) and/or ESS per target density gradient evaluation (ESS/grad). All results are given in minimum ESS over all dimensions unless otherwise noted. In terms of these performance measures, larger numbers are better.

Here we provide brief descriptions of the datasets used in our experiments:

Ill Conditioned Gaussian: a 50-dimensional ill-conditioned Gaussian task described in [10]; a Gaussian with diagonal covariance matrix with log-linearly distributed entries between $\left[10^{-2},10^{2}\right]$.

Strongly correlated Gaussian: a 2-dimensional Gaussian with variance $\left[10^{2},10^{-1}\right]$ rotated by $\frac{\pi }{4}$; same as in [10].

Funnel distribution: The density function is $p_{funnel}\left(x\right)=N\left(x_{0};0,\sigma^{2}\right)$$N(x_{1:n};0,I\exp\left(-2x_{0}\right))$. This is a challenging distribution because the spatial scale of $x_{1:n}$ varies drastically depending on the value of $x_{0}$. This geometry causes problems to adaptation algorithms that rely on a spatial scale. An important detail is that earlier work, such as [35] used σ=3, while some recent works used σ=1. We ran experiments with σ=1 for comparison with recent techniques and also used our method on a 20 dimensional funnel distribution with σ=3. We denote the two variants by Funnel-1 and Funnel-3.

Bayesian logistic regression: we follow the setup in [34] and used the German, Heart and Australian datasets from the UCI data registry.

In Figure 2, we compare our method with HMC on the 20d Funnel-3 distribution. As discussed in [35], the stepsize of HMC needs to be manually tuned down to allow traveling into the neck of the funnel, otherwise the sampling process will be biased. We thus tuned the stepsize of HMC to be the largest that still allows traveling into the neck. Each HMC proposal was set to use the same number of gradient steps as each proposal of the trained sampler. As can be seen, the samples proposed by our method travel significantly further than the HMC samples. Our method achieved 0.256 (ESS/MH), compared to 0.0079 (ESS/MH) with HMC.

As a demonstration we provide a visualization of the resulting chain of samples in Figure 2 and the learned proposal distributions in Appendix Figure A2. The energy value for the neck of the funnel can be very different than for the base, which makes it hard for methods such as HMC to mix between them [35]. In contrast, our model can produce very asymmetric $q(x^{\prime }|x)$ and $q(x|x^{\prime })$, making mixing between different energy levels possible.

Performances on other synthetic datasets and the Bayesian logistic regression are shown in Table 1. With all these datasets our method outperformed previous neural network-based MCMC approaches by significant margins. Our model also outperformed gradMALA [14], which uses the same objective but only use linear adaptive parameters. The experiments used various parameter settings, as detailed in Section A.2. Results of other models were adopted or converted from numbers reported in the original papers. The Appendix provides further experimental results, ablation studies, visualizations and details on the implementation of the model.

### 5.2. Training a Convergent Deep Energy-Based Model

A very challenging application of the MCMC method is training a deep energy-based model (EBM) of images [2,36,37]. We demonstrate stable training of a convergent EBM, and that the learned sampler achieves better proposal entropy early during training, as well as better sample quality at convergence, compared to the MALA algorithm. An added benefit is that, like in adaptive MALA, tuning the Langevin dynamics step size is no longer needed, instead, one only needs to specify a target accept rate. This contrasts earlier work using unadjusted Langevin dynamics, where step size needs to be carefully tuned [2].

Following [2], we used the Oxford flowers dataset of 8189 28 × 28 colored images. We dequantize the images to 5 bits by adding uniform noise and use logit transform [17]. Sampling was performed in the logit space with variant 2 of our sampler (without the *R* network; see Section A.1). During training, we used persistent contrastive divergence (PCD) [38] with a replay buffer size of 10,000. We alternated between training the sampler and updating samples for the EBM training. Each EBM training step uses 40 sampling steps, with a target accept rate of 0.6.

Figure 3 depicts samples from the trained EBM replay buffer, as well as samples from a 100,000 step sampling process—for demonstrating that in general the sampling sequence converges to a fixed low energy state. We also show that early during training, the proposal entropy of the learned sampler is higher than that of an adaptive MALA algorithm with the same accept rate target. Later during training, the proposal entropy is not significantly different (See Figure A3a). This is likely because the explorable volume around samples becomes too small for the learned sampler to make a difference. Additionally, we show that the model trained with the learned sampler achieved better sample quality by evaluating the Fréchet Inception Distance (FID) [39] between the replay buffer and ground truth data. A model trained with the learned sampler achieved 38.1 FID, while a model trained with MALA achieved 43.0 FID (evaluated at a late checkpoint, lower is better). We provide a plot that tracks the FID during training in Section A.3
Figure A3.

## 6. Discussion

In this paper we propose a gradient based neural network MCMC sampler with tractable proposal probability. The training is based on the entropy-based exploration speed objective. Thanks to an objective that explicitly encourages exploration, our method achieves better performance than previous neural network-based MCMC samplers on a variety of tasks, sometimes by an order magnitude. We also improved the FID of samples from the trained EBM from 41 to 38. Compared to the manifold HMC [35] methods, our model provides a more scalable alternative for mitigating unfavorable geometry in the target distribution.

There are many potential applications of our method beyond what is demonstrated in this paper—for example, training latent-variable models [40], latent sampling in GANs [41] and others applications outside machine learning, such as molecular dynamics simulations [1]. For future research, it would be interesting to investigate other architectures, such as an auto-regressive architecture, or alternative masking strategies to improve the expressiveness of the proposed model. Another promising direction could be to combine our technique with a neural transport MCMC.

Our proposed sampler provides more efficient exploration of the target distribution, as measured by the ESS results and proposal entropy. However, our method still has the limitation that the exploration is local. In EBM training, for example, as reported previously [2], the learned energy landscape is highly multi-modal with high energy barriers in between the minims. A sample proposal cannot cross those high barriers since it will result in high rejection probability. Our sampler achieves a small level of mixing as is visible in some sampling examples. However, our sampler, being a local algorithm, cannot explore different modes efficiently—it exhibits the same shortcoming in mixing as Langevin dynamics sampler.

## Figures and Tables

**Figure 1 entropy-23-00269-f001:**
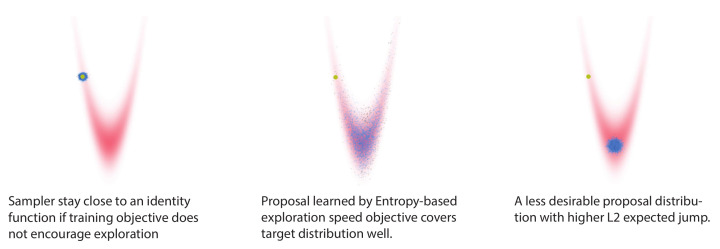
Illustration of learning for exploring a state space. Left panel: a Langevin sampler that has poor exploration. Middle panel: our proposed method—samples travel far within the target distribution. Right panel: a sampler with a higher L2 jump than ours—the exploration is still worse. In each panel, the yellow dot on the top left is the initial point *x*; blue and black dots are accepted and rejected samples, respectively.

**Figure 2 entropy-23-00269-f002:**
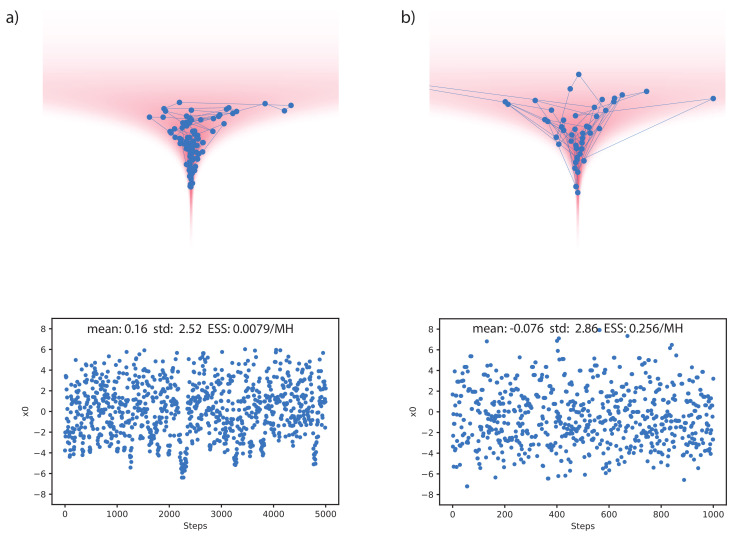
Comparison of our method with Hamiltonian Monte Carlo (HMC) on the 20d Funnel-3 distribution. (**a**) Chain and samples of $x_{0}$ (from neck to base direction) for HMC. (**b**) Same as (**a**) but for our learned sampler. Note, samples in (**a**) look significantly more correlated than those in (**b**), although they are plotted over a longer time scale.

**Figure 3 entropy-23-00269-f003:**
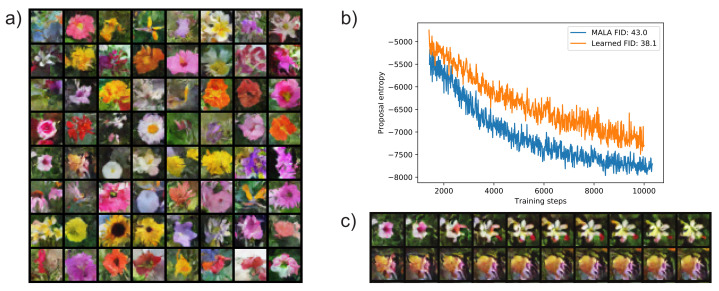
Training of the convergent energy-based model (EBM) with pixel space sampling. (**a**) Samples from replay buffer after training. (**b**) Proposal entropy of trained sampler vs. Metropolis-adjusted Langevin algorithm (MALA) early during training—note that the entropy of the learned sampler is significantly higher. (**c**) Samples from 100,000 sampling steps by the learned sampler, initialized at samples from replay buffer. Large transitions like the one in the first row are rare; this atypical example was selected for display.

**Table 1 entropy-23-00269-t001:** Performance comparisons. SCG: strongly correlated Gaussian. ICG: ill-conditioned Gaussian. German, Australian, Heart: Datasets for Bayesian logistic regression. ESS: effective sample size (a correlation measure).

Dataset (Measure)	L2HMC		Ours
50d ICG (ESS/MH)	0.783		**0.86**
2d SCG (ESS/MH)	0.497		**0.89**
50d ICG (ESS/grad)	$7.83\times 10^{-2}$		$2.15\times 10^{-1}$
2d SCG (ESS/grad)	$2.32\times 10^{-2}$		$2.2\times 10^{-1}$
**Dataset (Measure)**	**Neutra**		**Ours**
Funnel-1 $x_{0}$ (ESS/grad)	$8.9\times 10^{-3}$		$3.7\times 10^{-2}$
Funnel-1 $x_{1\cdots 99}$(ESS/grad)	$4.9\times 10^{-2}$		$7.2\times 10^{-2}$
**Dataset (Measure)**	**A-NICE-MC**	**NFLMC**	**Ours**
German (ESS/5k)	926.49	1176.8	**3150**
Australian (ESS/5k)	1015.75	1586.4	**2950**
Heart (ESS/5k)	1251.16	2000	**3600**

## Data Availability

Not applicable.

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
