# Peer review of "A Neural Network MCMC Sampler That Maximizes Proposal Entropy"

_entropy, 2021, doi:10.3390/e23030269_

Round 1
Reviewer 1 Report
The document is interesting and certainly engaged the authors to draft the
document. The structure of the paper is pleasant and the scientific and
methodological rigor is not lacking.
Perhaps the title hardly reflects the content of the paper, so if the authors thought it appropriate, I would extend the length of the title to make it more meaningful than the content of the paper as a whole.
The abstract is concise and qualitatively reports the most important results. However, no reference was made to the numerical results obtained. Reference should be made to this in the abstract to increase the reader's interest.
In the introduction reference was made to the dimensional analysis for HMC. Please associate the corresponding bibliographic reference.
The caption of figure 1 appears too large. I realize that the captions have to be self explanatory, but this seems too extensive. I ask the authors to reduce it as much as possible.
Section 2 begins by describing probability formulations directly in the text without numbering them. Please number all formulas in the text.
It is not clear how to arrive at (1) and (2). Please specify the individual passages that lead to their formulation.
The log determinant of N steps of transformation is (3). Why? Please, specify.
The reviewer is very appreciative of Section 4 which performed an in-depth study of related works. However, there is a strong emphasis on the possibility of uncertainty and / or imprecision in the usable data for which the neural approach used would require synergies of fuzzy and neuro-fuzzy models. It being understood that such applications could be the subject of future applications, the authors could mention this possibility in the text by citing some works related to this activity. In particular, I recommend the inclusion in the bibliography of the following works:
doi: 10.1155/2014/201243
The first work processes data affected by uncertainty and / or inaccuracy through the use of fuzzy techniques based on latest generation clustering
Figure 3 (b) is too small. Please enlarge it as much as possible.
Author Response
Thanks for the kind and informative review.
Based on your comments, we made the following adjustments:
We have added more highlights to our numerical results in the Abstract.
We have added references to step number scaling of various MC algorithms.
We have shortened the caption of figure 1 to make it more concise.
We have numbered formulas in section 2 that was mentioned later, and added the needed equation referencing.
We have made Figure 3 part b) larger by removing some samples from part c)
We have added the suggested reference on fuzzy clustering to related works.
We have added additional explanation to the model formulation section.
Thanks.
Reviewer 2 Report
Paper deals with important task. Authors proposed ANN-based approach for optimization proposal entropy for adapting the proposal to distributions of any shape.
Paper has scientific novelty and great practical value.
I am very like the structure of this paper. All additional information is on Apendixes. Its very good!
Paper is technically sound. Experimental section is good.
The proposed approach are logical.
Suggestions:
- It would be good to add point-by-point the main contributions in the end of the Introduction section.
- Related works section should be extended using non-iterative approaches for training. Authors should use tSGTM neural-like structure and its modiffications
- Authors should provide a short description on dataset used for modeling
- Authors should add all optimal parameters for all investigated methods
- Conclusion section should be extended using: 1) numerical results obtained in the paper; 2) limitations of the proposed approach; 3) prospects for the future research.
- A lot of references are outdated. Please fix it using 3-5 years old papers in high-impact journals.
- Authors used a lot of abreviations in the text of work. Please encrease Abbreviations section using all such issues
Author Response
Thanks for the kind and informative review.
Based on your comments, we made the following adjustments:
We have modified the last part of the introduction to include a point by point layout of our contributions.
We have updated the related works section to include suggested reference on SGTM neural-like structure.
We have updated the discussion section to include highlights of numerical results, as well as better discussion of limitations of our result. We already have a discussion of future works.
We have updated the abbreviation list to include as much mentioned abbreviations as possible.
We would also like to note that:
Description of all datasets are in Section 5.
Parameter settings for experiments with the proposed method is listed in Appendix Table 2. For other techniques, we used numbers reported in the original papers, and we refer readers to the original papers for parameter settings.
Thank you very much.